# κB-Ras and Ral GTPases regulate acinar to ductal metaplasia during pancreatic adenocarcinoma development and pancreatitis

Stephanie Beel [1], Lina Kolloch[1], Lisa H. Apken[1], Lara Jürgens[1], Andrea Bolle[1], Nadine Sudhof[1], Sankar Ghosh[2], Eva Wardelmann [3], Michael Meistererns[1], Konrad Steinestel[3,4] & Andrea Oeckinghaus [1]✉

Pancreatic ductal adenocarcinoma (PDAC) is associated with high mortality and therapy resistance. Here, we show that low expression of κB-Ras GTPases is frequently detected in PDAC and correlates with higher histologic grade. In a model of KRas$^{G12D}$-driven PDAC, loss of κB-Ras accelerates tumour development and shortens median survival. κB-Ras deficiency promotes acinar-to-ductal metaplasia (ADM) during tumour initiation as well as tumour progression through intrinsic effects on proliferation and invasion. κB-Ras proteins are also required for acinar regeneration after pancreatitis, demonstrating a general role in control of plasticity. Molecularly, upregulation of Ral GTPase activity and Sox9 expression underlies the observed phenotypes, identifying a previously unrecognized function of Ral signalling in ADM. Our results provide evidence for a tumour suppressive role of κB-Ras proteins and highlight low κB-Ras levels and consequent loss of Ral control as risk factors, thus emphasizing the necessity for therapeutic options that allow interference with Ral-driven signalling.

[1] Institute of Molecular Tumorbiology, Faculty of Medicine, University Münster, Münster, Germany. [2] Department of Microbiology & Immunology, College of Physicians & Surgeons, Columbia University, New York, NY, USA. [3] Gerhard-Domagk-Institute of Pathology, Faculty of Medicine, University Münster, Münster, Germany. [4] Institute of Pathology and Molecular Pathology, Bundeswehrkrankenhaus Ulm, Ulm, Germany. ✉email: oeckinga@ukmuenster.de

Pancreatic cancer is highly lethal, and due to its late diagnosis, rapid metastasis and its resistance to chemotherapy, remains difficult to treat[1,2]. The pancreas is a mixed gland: the exocrine pancreas consists of acinar cells that secrete digestive enzymes and ducts through which the pancreatic juice is transported. Langerhans islets carry out the endocrine functions by secreting hormones to control blood glucose levels. Most pancreatic tumours arise in the exocrine tissue with pancreatic ductal adenocarcinoma (PDAC) accounting for 95% of tumours. PDAC generally develops from pancreatic intraepithelial neoplasias (PanINs), the most common preneoplastic lesions in humans[3]. Progression from low-grade (PanIN1) over high-grade (PanIN3) lesions to PDAC occurs with successive accumulation of mutations. Activation of the small GTPase KRas (rat sarcoma) is considered the critical initial event, and activating KRas mutations are found in 95% of all PDACs[4]. In mice, specific expression of KRas$^{G12D}$ in the pancreas recapitulates the histological features of human disease with mice developing PanINs, but without additional genetic alterations only rarely invasive carcinomas[5].

Ras family GTPases function by cycling between a GDP-bound OFF and a GTP-bound ON state. This cycle is regulated by guanine nucleotide exchange factors (GEFs), which promote exchange of GDP to GTP, and by GTPase-activating proteins (GAPs), which stimulate GTP hydrolysis and return to the OFF state[6]. Ras activation triggers three major downstream effector pathways that contribute to transformed growth: the Raf/MAP-Kinase (rapidly accelerated fibrosarcoma/mitogen-activated protein kinase) pathway, the PI3K (phosphatidylinositol-3-kinase)/ Akt pathway and the Ral (Ras-like) GTPase signalling pathway (Fig. 1a)[7]. Recent findings suggest a complex role of PI3K–Pdk1 signalling in PDAC. Constitutively active PI3K (PI3KCA$^{H1047R}$) has been shown to mimic KRas$^{G12D}$-driven

tumorigenesis, while Pdk1 (3-phosphoinositide-dependent protein kinase-1) deletion in the KRas$^{G12D}$-expressing pancreas blocked PanIN and PDAC development[8,9]. These findings are supported by the fact that PTEN (phosphatase and tensin homologue) deletion in combination with KRas$^{G12D}$ expression strongly promoted pancreatic carcinogenesis[10–12]. Notably, another study reported conditional expression of oncogenic BRaf$^{V600E}$ (v-Raf murine sarcoma viral oncogene homologue B) but not PI3KCA$^{H1047R}$ to induce PDAC development; however, the contribution of BRaf to KRas-driven tumorigenesis has not been directly investigated[13]. So far, no in vivo models have addressed the role of Ral GTPases in PDAC, but RalA and RalB are consistently hyperactivated across pancreatic cancer cell lines and tumour samples, and Ral activation mediates AIP and invasion of PDAC cell lines[14–16].

The pancreas is a highly plastic organ that assures organ functionality under changing physiological situations and recovery from stress[17]. As part of this plasticity, acinar cells are able to transcriptionally and morphologically revert to cells with progenitor- and duct-like characteristics, a process called acinar-to-ductal metaplasia (ADM). These cells can then proliferate and replenish the damaged organ. Interestingly, accumulating evidence from studies with KRas$^{G12D}$-expressing mice suggests that ADM is triggered by mutated KRas and precedes the development of PanINs[18–22]. ADM also occurs in response to pancreatitis. Here, ADM is generally transient, and acinar cells revert to their differentiated state within days to weeks. In the presence of an additional genetic insult or continued cellular stress, ADM is rendered irreversible, and progression to PanIN becomes inevitable[18,19].

Reduced mRNA expression of the GTPases κB-Ras1 (NF-κB inhibitor interacting Ras-like protein 1, *NKIRAS1*) and

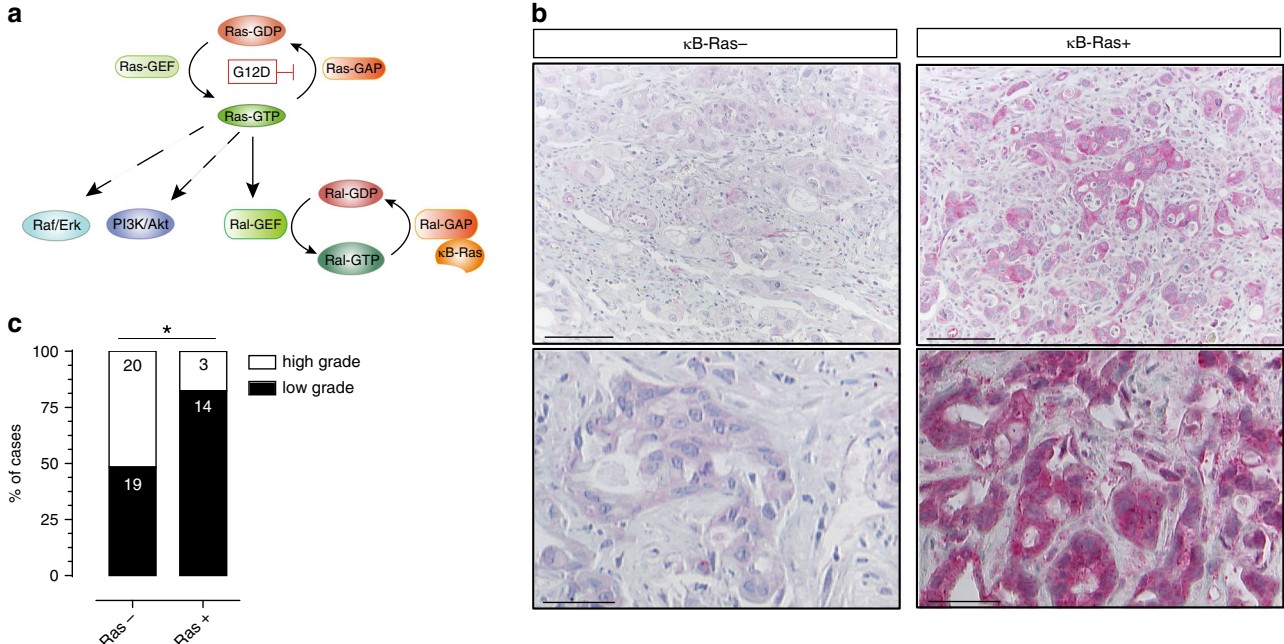

**Fig. 1 Low κB-Ras expression in PDAC patient samples. a** Signalling scheme of Ras effector pathways and control of Ral activity through κB-Ras proteins. G12D mutation in Ras leads to GTP-locked and thus constitutively active Ras. Ras-GTP can trigger activity of the MAPK, the PI3K/Akt and the Ral pathway. κB-Ras:Ral-GAP complexes limit Ral GTPase activity through increasing the low intrinsic GTP hydrolysis rate of RalA and RalB, and thus promoting the GDP-bound OFF state. Raf Rapidly Acc fibrosarcoma kinase, Erk extracellular signal-regulated kinase, PI3K phosphatidylinositol-3-kinase, GAP GTPase-activating protein, GEF guanine nucleotide exchange factor. **b** Immunohistochemical staining of human PDAC samples showing weak (left) and strong (right) κB-Ras expression. Upper panel scale bar: 100 µm; lower panel scale bar: 40 µm. **c** Statistical analysis revealed a significant association between histological tumour grade and expression levels of κB-Ras (Fisher's exact test, $p = 0.0216$). Numbers of patients are given on the graph (high grade $n = 23$ and low grade $n = 33$ patients).

κB-Ras2 (NF-κB inhibitor interacting Ras-like protein 1, NKIRAS2) has been detected in several solid cancers, including PDAC[23]. κB-Ras1 and κB-Ras2 are closely related isoforms, which share high homology with KRas, but also harbour distinctive features, such as presumably inactivating amino acid substitutions at positions critical for intrinsic GTPase activity and lack of a C-terminal farnesylation site[24]. κB-Ras proteins have been discovered through their interaction with the NF-κB inhibitor protein IκBβ and shown to function as negative regulators of NF-κB activity in macrophages to limit inflammatory responses in vivo[23,24]. Inhibition of IκBβ degradation and nuclear translocation or phosphorylation of the NF-κB subunit p65 have been attributed to κB-Ras proteins in this context[24–27]. In addition, κB-Ras proteins are associated with the Ral GTPase-activating protein complexes Ral-GAP1 and Ral-GAP2, which harbour GAP activity for the Ral GTPases[28–30]. Interestingly, κB-Ras binding to Ral-GAP is required to maintain GAP function and thus restrict Ral activation[23] (Fig. 1a). Both NF-κB and Ral pathways are closely connected to tumorigenesis. NF-κB family transcription factors can inhibit apoptosis or support invasion, and aberrant NF-κB activity is detected in many solid tumours, including PDAC[31–33]. The regulation of Ral and/or NF-κB by κB-Ras could therefore impact tumorigenesis[34], but whether deregulation of κB-Ras is relevant for tumour development in vivo has remained elusive.

Here, we show that low expression of κB-Ras frequently occurs in PDAC patients and correlates with the presence of higher-grade carcinomas. We demonstrate that loss of κB-Ras proteins cooperates with oncogenic Ras to promote development of invasive carcinoma in mice, resulting in dramatically shortened median survival. We find that κB-Ras proteins control both tumour initiation through regulation of ADM and tumour progression through intrinsic effects on cell proliferation and invasion. κB-Ras loss also leads to impaired regeneration of the exocrine tissue and persistent ADM after acute pancreatitis independently of KRas$^{G12D}$ expression, demonstrating a general role in acinar plasticity. Mechanistically, enhanced Ral GTPase activity drives these phenotypes. These findings describe a previously unrecognised function of Ral GTPase signalling in ADM regulation, and identify deregulation of κB-Ras levels and consequent loss of Ral control as a risk factor for pancreatic cancer development.

## Results

**Low κB-Ras levels in human PDAC patients**. Analysing publicly available data sets, we had previously recognised that κB-Ras mRNA levels are reduced in different solid cancers, including glioma, glioblastoma, invasive breast carcinoma, lung adeno-carcinoma and PDAC[23]. To address the relevance of these observations for human disease, we now examined a tissue microarray containing 81 samples from 61 PDAC patients for κB-Ras expression levels via immunohistochemistry (Supplementary Table 1). In total, 69.1% of cases showed absent or weak expression of κB-Ras proteins, confirming frequent reduction of κB-Ras levels in this tumour type (Fig. 1b, c). Notably, low κB-Ras levels were positively correlated with a higher histological tumour grade (Fig. 1c), arguing for a tumour-suppressive role of κB-Ras proteins that affects PDAC.

**κB-Ras deficiency does not affect pancreas development**. While κB-Ras1 and κB-Ras2 single-knockout mice are viable and phenotypically inconspicuous, concurrent knockout of both κB-Ras proteins results in perinatal lethality[23]. To analyse the role of κB-Ras proteins in the pancreas, we therefore generated a conditional κB-Ras2-knockout allele (*NKIRAS2$^{fl}$*) (Supplementary Fig. 1a, b). This mouse was then crossed with conventional κB-Ras1 and κB-

Ras2 knockout as well as Pdx1-Cre mice[35] to obtain animals with pancreas-specific deletion of κB-Ras2 without (*NKIRAS2$^{fl/fl}$ Pdx1-Cre$^{+}$* or *NKIRAS2$^{fl/−}$ Pdx1-Cre$^{+}$*, hereafter referred to as pancreatic 2SKO (p2SKO)) or with a concomitant general deficiency of κB-Ras1 (*NKIRAS1$^{−/−}$ NKIRAS2$^{fl/−}$ Pdx1-Cre$^{+}$* or *NKIRAS1$^{−/−}$ NKIRAS2$^{fl/fl}$ Pdx1-Cre$^{+}$*, pDKO mice) (Supplementary Fig. 1c). Since the Pdx1 promoter becomes active in the pre-pancreatic endoderm, Cre expression is driven in the complete pancreatic epithelium, including endocrine and exocrine cell types[36]. Successful loss of κB-Ras2 expression was confirmed in whole pancreas samples by RT-qPCR and immunoblot, being equally efficient whether one or two floxed alleles had to be deleted (Fig. 2a, b). p2SKO and pDKO mice were born at the expected ratios, were viable and grew to adulthood without phenotypic abnormalities. Both lines and littermate κB-Ras1 single-knockout mice (*NKIRAS1$^{−/−}$ NKIRAS2$^{+/+}$ Pdx1-Cre$^{+}$*, 1SKO) exhibited normal pancreatic cytoarchitecture and differentiation (Fig. 2c). Consistent with their healthy appearance, adult pDKO mice had normal responses in glucose tolerance tests and normal serum amylase levels (Fig. 2d, e). Thus, κB-Ras proteins are dispensable for normal pancreatic development and function. No tumour development was observed in pDKO mice up to 43 weeks of age, suggesting that κB-Ras deficiency alone is not sufficient to initiate tumorigenesis.

**κB-Ras deficiency promotes pancreatic tumorigenesis**. To ask whether κB-Ras proteins are relevant for K-Ras$^{G12D}$-driven pancreatic cancer, we crossed wild-type, 1SKO, p2SKO and pDKO animals with mice carrying the lox-STOP-lox (LSL) KRas$^{G12D}$ allele, which allows for conditional expression of oncogenic KRas$^{G12D}$[5]. For our analyses, we generated cohorts of KRas$^{G12D}$-expressing wild-type (*NKIRAS1$^{+/+}$ NKIRAS2$^{+/+}$ KRas$^{LSL-G12D/+}$ Pdx1-Cre$^{+}$*, hereafter referred to as WT/R), 1SKO (*NKIRAS1$^{−/−}$ NKIRAS2$^{+/+}$ KRas$^{LSL-G12D/+}$ Pdx1-Cre$^{+}$*, 1SKO/R), p2SKO (*NKIRAS1$^{+/+}$ NKIRAS2$^{fl/−}$ (or NKIRAS2$^{fl/fl}$) KRas$^{LSL-G12D/+}$ Pdx1-Cre$^{+}$*, p2SKO/R) and pDKO mice (*NKIRAS1$^{−/−}$ NKIRAS2$^{fl/−}$ (or NKIRAS2$^{fl/fl}$) KRas$^{LSL-G12D/+}$ Pdx1-Cre$^{+}$*, pDKO/R) (Supplementary Fig. 2a). All mice were born at expected ratios. However, pDKO/R mice presented with reduced weight gain as early as 3–4 weeks after birth, while WT/R, 1SKO/R and p2SKO/R mice were visually indistinguishable from WT mice. pDKO/R mice developed cachexia and abdominal distension around 8–12 weeks and became moribund between 10 and 18 weeks with a median survival of 14 weeks (Fig. 3a). Due to this early lethality, we examined histological changes of the pancreata at an age of 4 and 10 weeks. All WT/R mice presented well-preserved pancreatic lobules and islets of Langerhans along with normal ducts or only mild pancreatic intraepithelial neoplasia (PanIN low grade) at 4 and 10 weeks of age (Fig. 3b, c; Table 1). Strikingly, the combination of KRas$^{G12D}$ expression with complete κB-Ras deficiency (pDKO/R) resulted in the occurrence of carcinoma with invasive features already at 4 weeks, which continued to progress with age (Fig. 3b, c; Table 1). Tumour cells in pDKO/R pancreata expressed the ductal cell marker cytokeratin 19 (CK19) and mucin, characterising the observed tumours as PDAC (Fig. 3d–f). In line, pDKO/R tumours were accompanied by severe tissue desmoplasia and inflammation as determined by infiltration of CD45+ cells and expression of inflammatory cytokines, both hallmarks of PDAC (Fig. 3e, g, h; Supplementary Fig. 2b). Despite the fast development of PDAC, no metastases were found in the liver or lungs of pDKO/R mice during necropsy. Notably, no hyperactivation of KRas could be detected in the pancreata of pDKO/R mice in comparison with 1SKO/R littermates (Fig. 3i). Also, activation of MAPK and PI3K signalling was not grossly altered by κB-Ras deletion

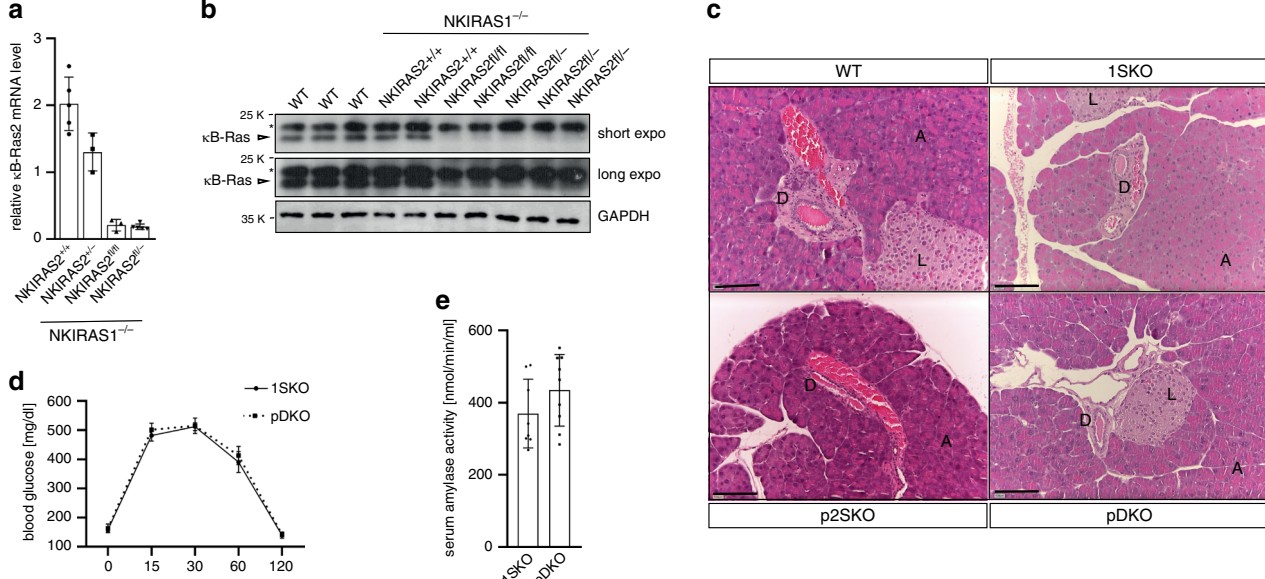

**Fig. 2 Normal development and function of the κB-Ras-deficient pancreas. a** RNA was prepared from mice with the indicated *NKIRAS2* genotypes (all *NKIRAS1*$^{-/-}$) and κB-Ras2 mRNA level analysed via RT-qPCR. Data are presented as mean values ± SD. $n = 5$ (*NKIRAS2*$^{+/+}$, *NKIRAS2*$^{fl/fl}$) or $n = 3$ (*NKIRAS2*$^{fl/-}$, *NKIRAS2*$^{+/-}$) biologically independent animals. **b** Protein extracts were prepared from pancreata of mice with the indicated *NKIRAS2* genotypes (all *NKIRAS1*$^{-/-}$) or WT mice and analysed by immunoblot for total κB-Ras protein level with GAPDH as control. Asterisk marks an unspecific band detected by the κB-Ras antibody. **c** Haematoxylin/eosin staining of paraffin sections of pancreata from 15- to 22-week-old WT, 1SKO, p2SKO and pDKO mice. A: acinar tissue, D: duct, L: islet of Langerhans. Scale bar: 100 μm. **d** 12-week-old pDKO and 1SKO littermate control animals were intraperitoneally injected with 2 g/kg glucose and blood glucose levels determined. Data are presented as mean values ± SD. $n = 9$ individual mice per genotype. **e** Amylase levels were determined in blood serum from 12-week-old pDKO and 1SKO mice. Data are presented as mean values ± SD. 1SKO: $n = 9$ and pDKO: $n = 9$ individual mice. Source data are provided as a source data file.

(Supplementary Fig. 2c). 1SKO/R and p2SKO/R mice revealed partial phenotypes. The percentage of 1SKO/R mice harbouring low-grade PanINs was not significantly different from that of WT/R animals; however, PanIN lesions tended to be more widespread (Fig. 3c). In the p2SKO/R group, the percentage of animals with low-grade PanIN at 4 weeks was slightly higher than in the 1SKO/R and WT/R cohorts, and at an age of 10 weeks, some p2SKO/R mice already presented with adenocarcinoma with invasive features and surrounding high-grade PanIN (Fig. 3b, c; Table 1). These results demonstrate that loss of κB-Ras dramatically decreases the latency of tumour onset providing genetic evidence for a role of κB-Ras in counteracting KRas$^{G12D}$-driven pancreatic cancer. Our characterisation further suggests a more prominent but still redundant role of κB-Ras2 over κB-Ras1 in the murine model system.

**κB-Ras controls Ral activity and tumour cell proliferation**. We noted that tumorigenic pDKO/R ductal cells had a higher proliferation index than 1SKO/R lesions when analysed by Ki67 staining (Fig. 4a, b). However, this might reflect the differential tumour progression in 1SKO/R and pDKO/R animals as positive Ki67 staining has been shown to increase with lesion grade[37,38]. To identify molecular alterations in primary ductal tumour cells, we first examined the activation status of NF-κB and Ral signalling via fluorescence-activated cell sorting (FACS) staining for phospho-NF-κB/p65 or phospho-TBK1 (TANK-binding kinase-1) as readouts for the respective pathways. Ductal cells were identified by surface staining with the MIC-1C3 antibody, which had previously been characterised to mark surface antigens of pancreatic duct cells[39]. In comparison with 1SKO/R animals, we detected an enhancement of TBK1 phosphorylation in pDKO/R ductal cells, suggesting increased Ral activity.

Phosphorylation of p65 was not significantly altered with high variability between animals (Fig. 4c, d; Supplementary Fig. 3a). To further investigate this finding, we generated pancreatic ductal cell lines (PDCs) from three pDKO/R and three 1SKO/R sex-matched littermate mice. Expression levels of ductal, acinar and islet markers confirmed identity of the obtained cells and immunoblot the effective deletion of κB-Ras (Fig. 4e, Supplementary Fig. 3b, c). No differences in NF-κB DNA binding under resting conditions, or when PDCs were challenged with PDAC-relevant cytokines, such as interleukins 1α and β (IL-1α and IL-1β) or tumour necrosis factor (TNF), were observed in electromobility shift assays (EMSA). We also stimulated cells with lipopolysaccharide (LPS) as we had previously observed differences in NF-κB activation in macrophages upon this stimulus[23], but again found no differences between 1SKO/R and pDKO/R PDCs (Fig. 4f, g, Supplementary Fig. 3d, e). As κB-Ras proteins exert their NF-κB-regulating function through IκBβ, we additionally tested NF-κB binding to a TNF promoter sequence that is regulated in an IκBβ-dependent manner. Also here, no difference in NF-κB DNA binding could be detected, suggesting that κB-Ras might not play the same role in NF-κB regulation in epithelial cells as in macrophages (Fig. 4f).

We next tested the effect of κB-Ras deletion on Ral-GTP levels in PDCs through GST-Sec5 pulldown followed by immunoblot. Despite the presence of oncogenic KRas$^{G12D}$, which constitutively drives Ral activity via Ral-GEFs[15], κB-Ras deletion led to a significant enhancement of Ral-GTP levels (Fig. 4h, Supplementary Fig. 3c). Indeed, Ral-GTP levels were increased so strongly that EGF stimulation was not able to further induce Ral activity. Stable re-expression of both κB-Ras proteins could reduce Ral activity and restore sensitivity to EGF stimulation, demonstrating that this effect is indeed mediated by loss of κB-Ras and not due

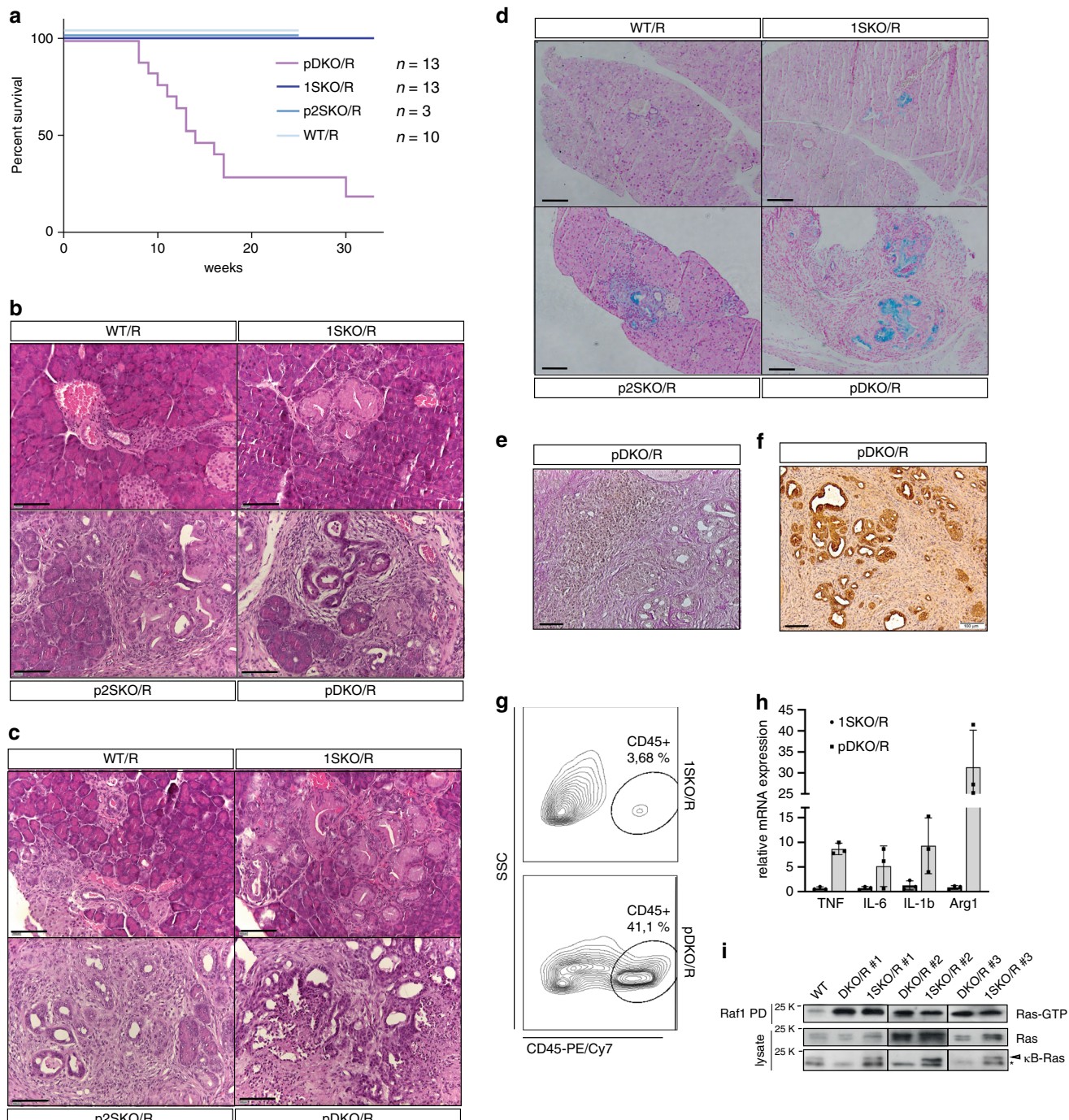

**Fig. 3 κB-Ras deficiency cooperates with oncogenic KRas to promote invasive pancreatic adenocarcinoma. a** Kaplan–Meier plot of WT/R (*n* = 10), 1SKO/R (*n* = 13), p2SKO/R (*n* = 3) and pDKO/R (*n* = 13) mice showing time until signs of illness necessitated euthanasia (Mantel–Cox analysis, *p* = 0.0003 for pDKO/R). **b, c** Haematoxylin/eosin-stained paraffin sections of pancreata from 4-week-old **b** or 10-week-old **c** WT/R, 1SKO/R, p2SKO/R and pDKO/R mice. Scale bar: 100 μm. **d** Alcian Blue/Nuclear Red staining of pancreata from 10-week-old WT/R, 1SKO/R, p2SKO/R and pDKO/R mice. Scale bar: 100 μm. **e** PAS (brown) and Elastica (red) staining of pDKO/R pancreas (10 weeks) revealing mucin production and desmoplasia, respectively. Scale bar: 100 μm. **f** Immunohistochemical staining for cytokeratin 19 (CK19) of pancreatic section of a 10-week-old pDKO/R mouse. Scale bar: 100 μm. **g** Representative FACS analysis for leucocytes (CD45+) in pDKO/R versus 1SKO/R pancreata at 10 weeks. **h** RT-qPCR analysis of whole pancreas mRNA. Data are presented as mean values ± SD. *n* = 3 individual mice. **i** Pancreata of WT, pDKO/R and 1SKO/R mice were lysed and subjected to GST-Raf1 pulldown. Ras-GTP, total Ras and κB-Ras protein levels were analysed by immunoblot. Source data are provided as a Source data file.

to differences in the state of cellular transformation (Fig. 4i). We then overexpressed the fast-exchanging RalA mutant F39L in 1SKO/R and pDKO/R cells, which has previously been demonstrated to be independent of GEF but still controlled by GAP function[40]. RalA F39L-GTP levels were significantly enhanced in pDKO/R compared with 1SKO/R cells (Fig. 4j). Furthermore, expression of the dominant-negative RalA S28N mutant, which can inhibit Ral-GEF-dependent activation of Ral GTPases[41–43], did not change endogenous Ral-GTP levels in pDKO/R PDCs (Supplementary Fig. 3f). Together, these findings demonstrate

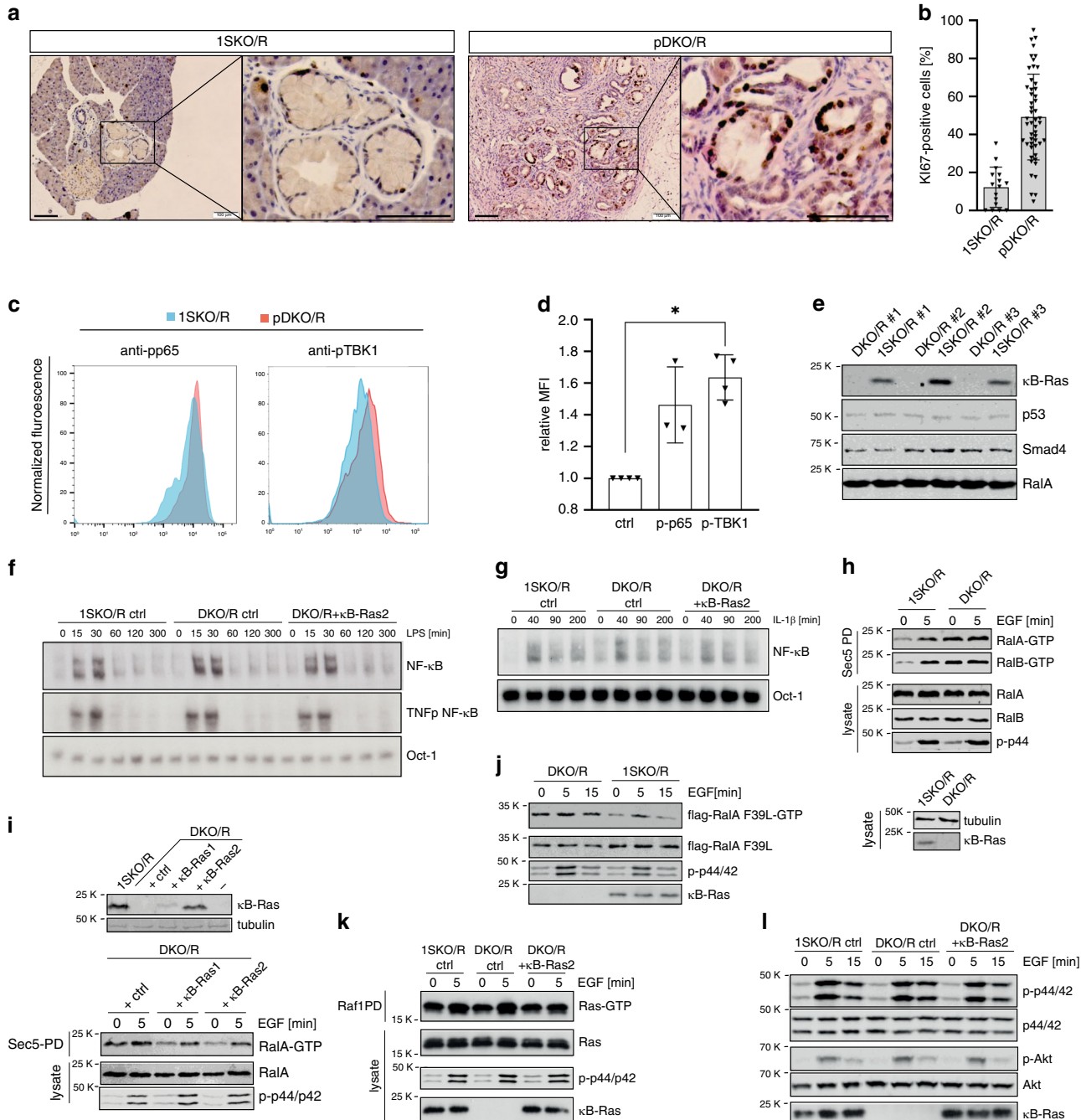

**Fig. 4 κB-Ras proteins limit Ral-GTP levels in pancreatic ductal tumour cells. a** Immunohistochemical staining for Ki67 of paraffin sections from 10-week-old pDKO/R and 1SKO/R mice (scale bar: 100 μm) and **b** quantification of Ki67-positive cells per ductal structure in the field of view. Data are presented as mean values ± SD. 1SKO/R: $n = 16$ and pDKO/R: $n = 51$ fields of view from 3 individual mice. Welch's t test $p = 2.83 \times 10^{-8}$. **c** FACS analysis of ductal (MIC-1C3⁺) cells from pancreata of 1SKO/R and DKO/R mice for phospho-p65 (S536) and phospho-TBK1 (S172) level with **d** quantification. Given is fold increase of mean fluorescence intensity (MFI) of pDKO/R relative to 1SKO/R cells. Data are presented as mean values ± SD. $n = 3$ (p-p65) or $n = 4$ (p-TBK1) independently performed analyses of individual mice. Two-sided t test, $p = 0.0473$. **e** Immunoblot of protein extracts from three pairs of generated PDC cell lines. **f** EMSA using whole-cell extracts from LPS-stimulated (1 μg/ml) 1SKO/R and DKO/R ctrl and DKO/R + κB-Ras2 (stable re-expression (see Fig. 4i)) cells. NF-κB = consensus-binding sequence probe, TNFp NF-κB = NF-κB binding sequence from TNF promotor. **g** EMSA using whole-cell extracts from IL-1β-stimulated (10 ng/ml) 1SKO/R, DKO/R ctrl and DKO/R + κB-Ras2 cells and the indicated probes. **h** Early-passage 1SKO/R and DKO/R PDCs were stimulated with EGF (100 ng/ml), lysed and subjected to GST-Sec5 PD. Ral-GTP and total Ral protein levels were analysed by immunoblot. **i** pDKO/R PDCs were retrovirally transduced to achieve stable re-expression of κB-Ras1 or κB-Ras2 (top). The resulting effect on RalA-GTP levels was investigated by GST-Sec5 pulldown and immunoblot. **j** pDKO/R and 1SKO/R PDCs were transfected with flag-RalA F39L, stimulated with EGF (100 ng/ml), lysed and subjected to GST-Sec5 PD. Flag-RalA F39L-GTP level was analysed by immunoblot. **k** 1SKO/R, DKO/R and DKO/R + κB-Ras2 cells were stimulated with EGF (100 ng/ml), lysed and subjected to GST-Raf1 PD. Ras-GTP level were analysed by immunoblot. **l** 1SKO/R, DKO/R and DKO + κB-Ras2 cells were stimulated with EGF (100 ng/ml) and lysed. Phospho-p44/42 and phospho-Akt S473 level were analysed by immunoblot. Source data are provided as a Source Data file.

| Genotype | Age | PanIN | PDAC | Median survival |
|----------|-----|-------|------|-----------------|
| WT/R | 4 W | 1/7 | 0/7 | >24 weeks |
| | 10 W | 5/7 | 0/7 | |
| 1SKO/R | 4 W | 3/6 | 0/6 | >32 weeks |
| | 10 W | 5/10 | 0/10 | |
| p2SKO/R | 4 W | 5/6 | 0/6 | >24 weeks |
| | 10 W | 7/7 | 5/7 | |
| pDKO/R | 4 W | 6/6 | 6/6 | 14 weeks |
| | 10 W | 8/8 | 8/8 | |

**Table 1 PanIN/PDAC incidences at ages 4 and 10 weeks and median survival for WT/R, 1SKO/R, p2SKO/R and pDKO/R mice.**

that κB-Ras deficiency results in increased Ral-GTP levels through loss of Ral-GAP activity in PDCs.

As in primary tissue, κB-Ras deficiency did not alter the KRas activation status of cells (Fig. 4k). Also, κB-Ras deficiency did not alter other Ras-driven pathways such as Raf/MAPK and PI3K/Akt signalling (Fig. 4l, Supplementary Fig. 3g). Also, Smad4 and p53 levels, as well as TGFβ (transforming growth factor β)-induced Smad signalling, were not affected by κB-Ras deficiency (Fig. 4e, Supplementary Fig. 3h). From this, we conclude that in our isolated PDCs, Ral but not NF-κB activity is deregulated upon κB-Ras deletion.

We then asked whether the increase in Ral-GTP levels upon κB-Ras deletion could indeed affect behaviour of KRas$^{G12D}$-expressing cells. While adherent growth was not uniformly altered between 1SKO/R and pDKO/R cell lines (Supplementary Fig. 4a), pDKO/R cells showed enhanced colony formation when grown under ultra-low-attachment conditions, revealing increased anchorage-independent proliferation (AIP). This capability could be diminished by re-expression of κB-Ras2 confirming that this effect is indeed due to κB-Ras deletion (Fig. 5a). Increased AIP of pDKO/R cells could be dampened by knockdown of Ral GTPases and treatment with the Ral inhibitor BQU57, demonstrating that this phenotype is mediated by enhanced Ral activity (Fig. 5b, c, Supplementary Fig. 4b). In line, pDKO/R cells were able to form tumours much quicker than 1SKO/R cells when grafted subcutaneously in NOD-Scid mice (Fig. 5d, Supplementary Fig. 4c). κB-Ras deficiency also led to enhanced invasiveness of PDCs as monitored by Matrigel Invasion Assays. Again, reintroduction of κB-Ras2 in pDKO/R PDCs reduced tumour growth and invasion (Fig. 5e).

To examine the relevance of these mechanisms in human cells, we knocked out both κB-Ras proteins in two patient-derived PDAC cell lines, Mia-Paca-2 and Panc1, using a CRISPR/Cas9 system (Supplementary Fig. 4d). Successful knockout was confirmed by both RT-qPCR and immunoblot (Fig. 5f, Supplementary Fig. 4e–g). Also, in these cell lines, loss of κB-Ras dramatically enhanced Ral-GTP levels while leaving NF-κB activity as well as PI3K/Akt and MAPK signalling unaltered (Fig. 5f–i, Supplementary Fig. 4h–j). As for murine PDCs, κB-Ras deficiency led to increased Ral-GTP levels through deregulation of Ral-GAP activity as demonstrated by increased GTP binding of RalA F39L and lack of effect of RalA S28N overexpression on levels of endogenous Ral-GTP (Fig. 5j, Supplementary Fig. 4k). κB-Ras-deficient human PDAC cells also showed increased AIP, an effect that could be dampened by knockdown of Ral GTPases (Fig. 5k, l). Similarly, invasion of DKO cells was increased demonstrating the functional impact of κB-Ras loss. As in the murine model system, single κB-Ras1- or κB-Ras2-knockout cells (1SKO and 2SKO) showed partial effects (Fig. 5m). In summary, these findings reveal that κB-Ras proteins limit Ral activation and thus AIP and tumorigenicity of fully transformed KRas$^{G12D}$-expressing PDAC cells.

**κB-Ras proteins control ADM during pancreatic tumorigenesis.** In addition to the high-proliferation index, pDKO/R pancreata revealed a fast loss of acinar tissue combined with an unusually large number of duct-like structures hinting at enhanced ADM (Fig. 6a). This observation was supported by RT-qPCR, showing increased expression of CK19 but decreased expression of amylase1 in whole pancreas of pDKO/R animals compared with 1SKO/R littermates (Supplementary Fig. 5a). We thus investigated a possible involvement of κB-Ras proteins in the ADM process. Indeed, we frequently detected co-expression of amylase1 and CK19 in cells forming duct-like structures in the pancreata of 4-week-old pDKO/R animals while their expression was mutually exclusive in 1SKO/R controls (Fig. 6b). To confirm this finding, FACS analysis using antibodies against surface antigens of acinar and ductal cells was performed[39]. In line with our RT-qPCR results, we found the number of cells expressing a ductal marker (MIC-1C3$^+$) generally increased in pancreata of pDKO/R mice (Fig. 6c; Supplementary Fig. 5b). Importantly, about 50% of the MIC-1C3$^+$ cells also expressed the acinar cell surface marker detected by the anti-MPx1 antibody[39]. To further confirm the specificity of these antibodies, the respective populations were sorted, and expression of CK19 and amylase1 analysed by RT-qPCR, revealing that MIC-1C3$^+$/MPx1$^+$ double-positive cells are indeed metaplastic (Supplementary Fig. 5c).

Despite the requirement of KRas mutation for PDAC, its expression alone is not sufficient to drive carcinogenesis beyond premalignancy, and ADM occurs with low penetrance and long latency. Secondary events, such as inflammatory or growth factor signalling and expression of the sex-determining region Y-box9 (Sox9) transcription factor in acinar cells, are needed for potent ADM and progression to PDAC[44]. Notably, EGFR signalling has been suggested to directly regulate Sox9 expression through NFATC1 and NFATC4 (nuclear factor of activated T cells' cytoplasmic 1 and 4)[45,46]. Interestingly, we find that loss of κB-Ras leads to upregulation of Sox9 levels in KRas$^{G12D}$-expressing acinar cells (Fig. 6d), suggesting that κB-Ras proteins might be involved in the control of Sox9 protein expression through EGFR/KRas signalling. Specificity of the Sox9 antibody was confirmed by Sox9 knockdown and subsequent IF analysis (Supplementary Fig. 5d, e). In summary, these results demonstrate that the absence of κB-Ras proteins promotes ADM in the KRas$^{G12D}$-expressing pancreas, accelerating the development and progression of lesions.

**κB-Ras proteins regulate ADM during acute pancreatitis.** ADM also occurs in response to pancreatitis where it is generally transient[47]. We next examined whether loss of κB-Ras would also alter ADM in response to cerulein-induced acute pancreatitis, independently of oncogenic KRas expression. pDKO and 1SKO littermates (no expression of KRas$^{G12D}$) were given seven hourly intraperitoneal injections of cerulein or saline, and pancreata were histologically analysed after different time intervals (Fig. 7a). Samples were scored in a blinded fashion based on the area affected by ADM (score 0: 0–2%; score 1: 2–50%; score 2: >50%) (Fig. 7b–d, Supplementary Fig. 6a, b). The induction of ADM was slightly, but not significantly enhanced in pDKO mice when compared with 1SKO controls. In line, measurement of serum amylase levels 9 h after the first injection confirmed equally efficient induction of tissue damage through cerulein in 1SKO and pDKO animals (Supplementary Fig. 6c). Robust acinar regeneration could be observed as early as 48 h after pancreatitis induction in 1SKO animals. At 21 days after cerulein injection, 1SKO pancreata were indistinguishable from those of control saline-injected animals. Additional loss of κB-Ras2 however led to dramatically compromised regeneration of the acinar

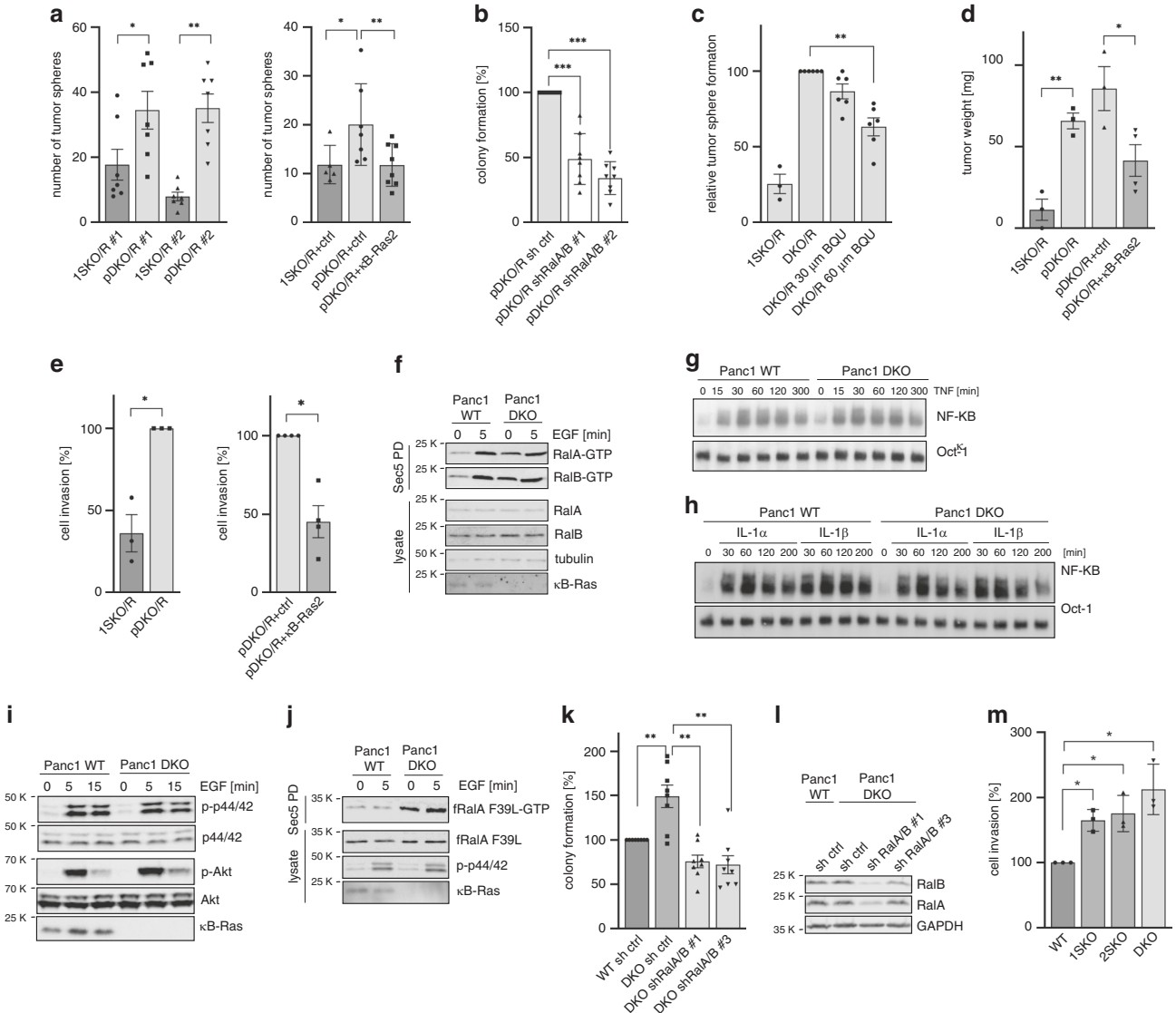

**Fig. 5 κB-Ras deficiency confers an intrinsic proliferation advantage to tumour cells.** If not stated otherwise, *n* number represents independent experiments, data are presented as mean values ± SEM, and a two-sided *t* test was performed. **a–c** PDC colony formation in ultra-low-attachment plates. **a** Left: two independently derived pairs of early-passage PDCs. *n* = 7; *\*p* = 0.04; *\*\*p* = 0.000074. Right: 1SKO/R and pDKO/R PDCs stably re-expressing κB-Ras2. *n* = 8; *\*p* = 0.0198; *\*\*p* = 0.0066. **b** Effect of knockdown of RalA/B in PDCs. *n* = 8; *p* = 0.0000036 (sh#1); *p* = 0.0000016 (sh#2). **c** Effect of BQU57. 1SKO: *n* = 3, DKO/R: *n* = 6; Welch's *t* test *p* = 0.0016. **d** NOD-Scid mice were grafted subcutaneously with PDCs and tumour weight was determined. *n* = 3 grafts from individual mice; *\*\*p* = 0.001; *\*p* = 0.028. **e** PDCs were seeded in matrigel-coated transwells and transmigration analysed. Left: *n* = 3; *p* = 0.03. Right: 1SKO/R and pDKO/R PDCs stably re-expressing κB-Ras2. *n* = 4; *p* = 0.013. **f** Panc1 WT and DKO cells were stimulated with EGF (100 ng/ml) and lysates subjected to GST-Sec5 pulldown followed by immunoblot. **g** EMSA using cell extracts from TNFα-stimulated (20 ng/ml) Panc1 WT and DKO cells. **h** EMSA using cell extracts from IL-1α- and IL-1β- (both 10 ng/ml) stimulated Panc1 WT and DKO cells. **i** Panc1 WT and DKO cells were stimulated with EGF (100 ng/ml) and p-p44/42 and p-Akt levels analysed by immunoblot. **j** Panc1 WT and DKO cells were transfected with Flag-RalA-F39L and subjected to GST-Sec5 pulldown and immunoblot. **k** Panc1 WT and DKO cells stably expressing RalA/B shRNAs were analysed for colony formation. *n* = 4. *\*\*p* = 0.0016 (WT sh ctrl-DKO sh ctrl); *\*\*p* = 0.0059 (DKO sh ctrl-DKO shRalA/B#1); *\*\*p* = 0.0015 (DKO sh ctrl-DKO shRalA/B#3). **l** Immunoblot of Ral levels in Panc1 DKO shRalA/B cells. **m** Mia-Paca2 WT and DKO cells were seeded in matrigel-coated transwells and transmigration analysed. Data are presented as mean values ± SD. *n* = 3. Welch's *t* test *\*p* = 0.0211 (WT-1SKO); *\*p* = 0.0425 (WT-2SKO); *\*p* = 0.0372 (WT-DKO). Source data are provided as a Source data file.

compartment, resulting in persistent metaplasia and acinar atrophy (Fig. 7b, d). Confirming the connection between κB-Ras and Sox9 regulation, we detected enhanced and prolonged Sox9 expression in acinar cells of pDKO animals (Fig. 7c, d, Supplementary Fig. 6a, b). Induction of MAPK and PI3K/Akt signalling after cerulein injection was equally efficient in 1SKO and pDKO mice and diminished during regeneration. Low-level activation of both pathways could still be observed in persistent ADM in

pDKO pancreata at 21 days after injection (Supplementary Fig. 7a, b). Whether this reflects the persistent metaplastic state or how it is linked mechanistically to κB-Ras deficiency, e.g. through crosstalk between signalling pathways, remains to be determined. These findings identify a role of κB-Ras proteins in acinar cell regeneration, and suggest that pathways that are controlled by κB-Ras are involved in regulating ADM during pancreatitis and cancer development.

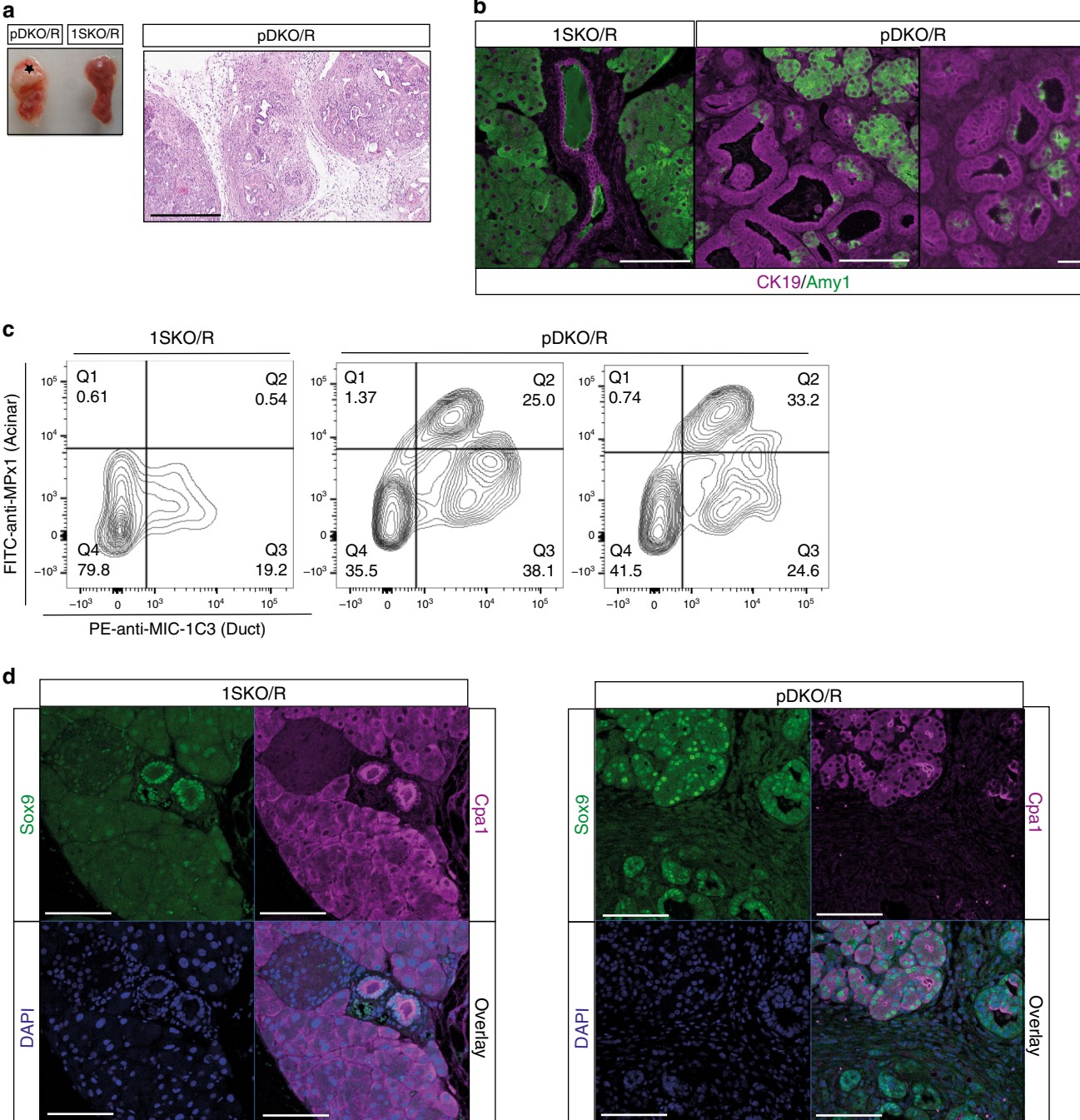

**Fig. 6 κB-Ras deficiency promotes acinar-to-ductal metaplasia and Sox9 expression. a** Resected tumours of 10-week-old 1SKO/R and pDKO/R mice (left) and exemplary haematoxylin/eosin staining of a section of 10-week-old pDKO/R mice (right). Scale bar: 500 μm. **b** Fluorescent IHC staining of CK19 (magenta) and amylase1 (green) in pancreatic sections of 1SKO/R and pDKO/R littermate mice (4 weeks old). Scale bar: 200 μm. **c** FACS analysis of pancreatic cell populations. Acinar cells were excluded from analysis based on FSC/SSC and residual cells analysed for ductal (MIC-1C3[+]) and acinar cell markers (MPx1[+]) by surface staining. **d** Fluorescent IHC staining of Cpa1 (magenta, acinar cell marker), Sox9 (green) and nuclei (DAPI, blue) in paraffin sections of 1SKO/R and pDKO/R littermate mice (4 weeks old). Scale bar: 200 μm.

**κB-Ras control of Ral activity is required to regulate ADM.** To understand the underlying molecular mechanisms of κB-Ras effects on ADM, we determined the activation status of Ral and NF-κB signalling in primary acinar cells in vivo. In agreement with our findings in ductal cells, phospho-TBK1 levels were clearly elevated in pDKO/R compared with 1SKO/R acinar cells, while phospho-p65 levels were not, suggesting that enhanced Ral-GTP levels might contribute to the observed effects on ADM.

FACS analysis also confirmed upregulation of Sox9 protein in pDKO/R acinar cells (Fig. 8a, b, Supplementary Fig. 8a). We next mimicked ADM in a 3D organoid assay system, in which primary acini are induced by TGFα (transforming growth factor α) stimulation to form duct-like structures. pDKO acini (no KRas[G12D] expression) were more potent in forming duct-like structures compared with 1SKO cells generated from sex-matched littermate mice (Fig. 8c). Importantly, treatment with

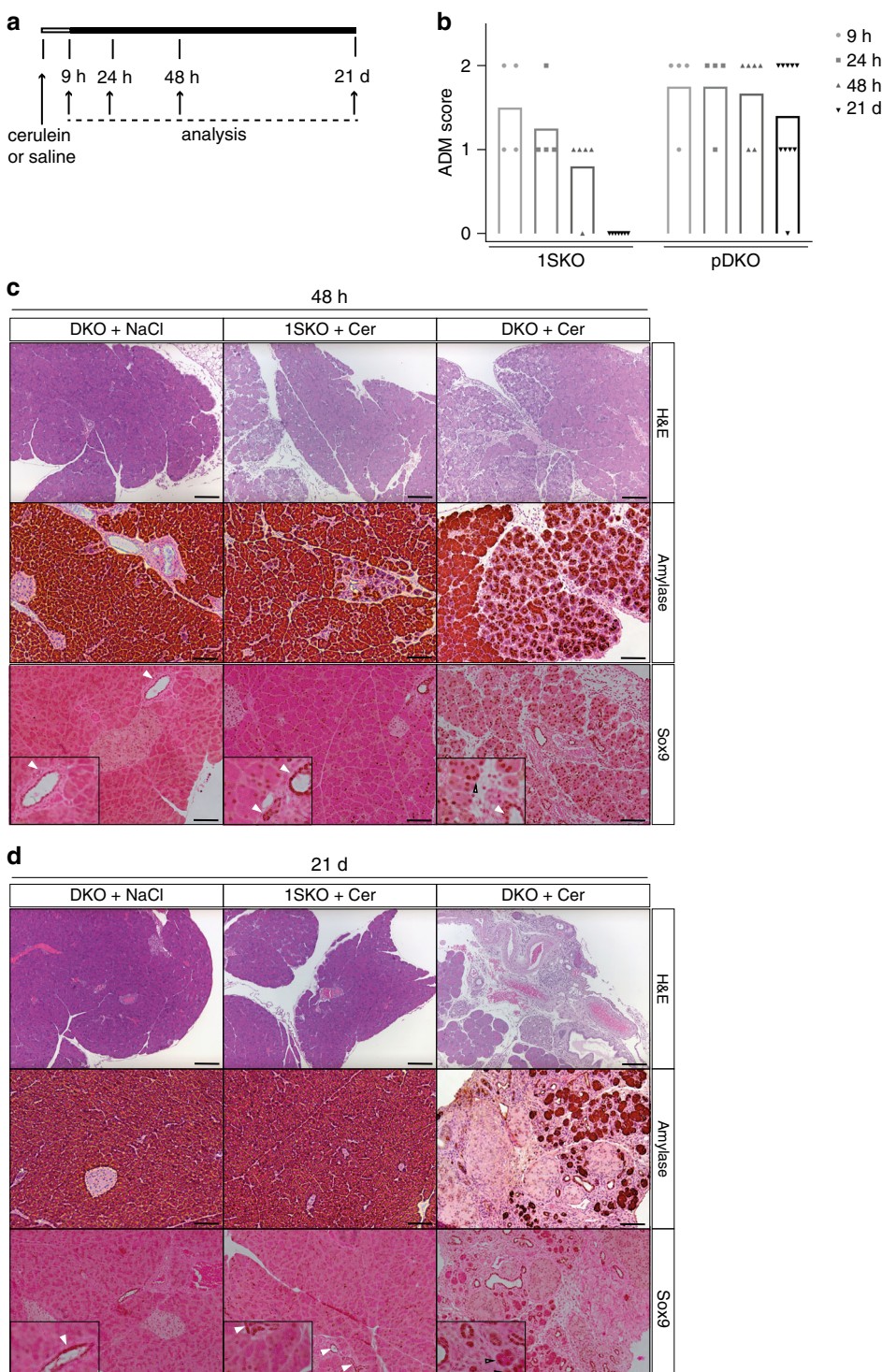

**Fig. 7 κB-Ras proteins are required for acinar regeneration after acute pancreatitis. a** Experimental set-up of acute pancreatitis model. **b** The percentage of pancreatic tissue affected by ADM after cerulein injections was scored in haematoxylin/eosin-stained sections (score 0: 0–2%; score 1: 2–50%; score 2: >50%). Data are presented as mean and single values. Data represent the collected results from two (9- and 24-h time points) or three (48-h and 21-d time points) independent experiments. **c, d** Sections of pancreata from mice with the indicated genotypes 48 h (**c**) and 21 d (**d**) after cerulein or saline injections were stained with haematoxylin/eosin or antibodies against amylase1 or Sox9. White arrows indicate Sox9 expression in ductal cells. Black arrows indicate Sox9 expression in acinar cells. Scale bar haematoxylin/eosin and amylase: 50 μm. Scale bar: Sox9: 100 μm.

BQU57 or knockdown of Ral GTPases reduced formation of duct-like structures by primary pDKO acini (Fig. 8d, e) without significant effects on cell viability (Supplementary Fig. 8b), demonstrating that Ral signalling indeed takes part in mediating ADM.

To confirm this role of κB-Ras proteins in an independent system, we generated a CRISPR/Cas9-mediated knockout of κB-Ras1 and κB-Ras2 in the rat acinar cell line AR42J (Fig. 8f, Supplementary Fig. S8c, d), and tested the effect of κB-Ras deletion in two clonal lines. Deletion of κB-Ras proteins led to

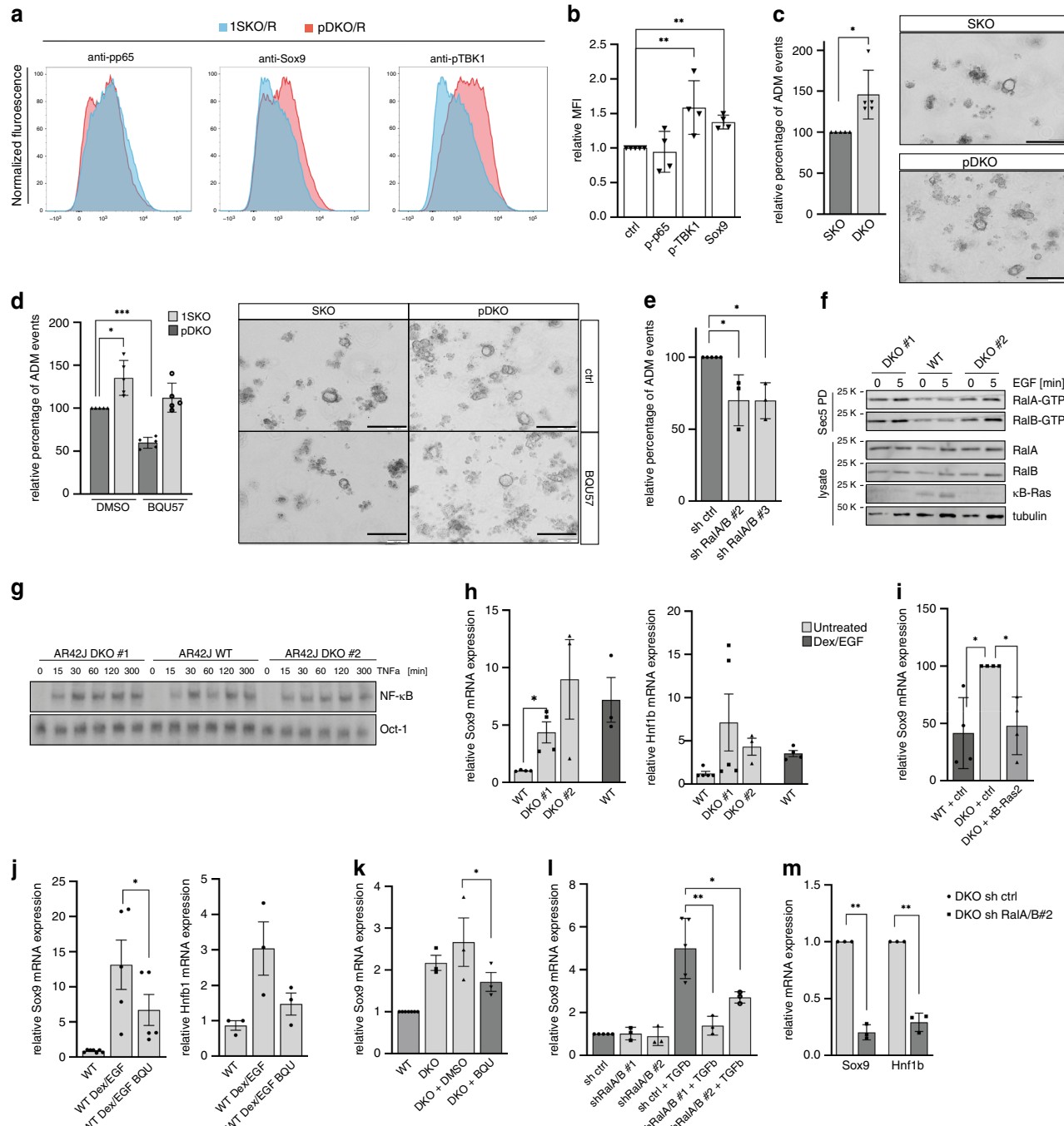

**Fig. 8 κB-Ras control of Ral activity is required to regulate Sox9 expression and counteract ADM.** If not stated otherwise, data are presented as mean values ± SD, and a two-sided *t* test was performed. **a** Primary acinar cells were analysed for phospho-p65 (S536), phospho-TBK1 (S172) and Sox9 levels via FACS and **b** mean fluorescence intensities (MFI) quantified. Given is fold increase of pDKO/R over 1SKO/R as mean values ± SD. *n* = 4 samples of individual mice. Welch's *t* test **p = 0.0048. **c–e** Primary acini were isolated from 1SKO and pDKO mice, and formation of ductal structures in collagen analysed. **c** *n* = 5. Welch's *t* test *p = 0.0265. Scale bar: 200 μm. **d** Primary acini were treated with 30 μM BQU57. *n* = 5. Welch's *t* test ***p = 0.0002; *p = 0.0179. **e** Primary acini were transduced with ctrl or RalA/B shRNAs. *n* = 3. One-sided *t* test p = 0.049 (sh#2); p = 0.029 (sh#3). Scale bar: 200 μm. **f** AR42J WT and DKO cells were stimulated with EGF (100 ng/ml) and subjected to GST-Sec5 pulldown. Ral-GTP levels were analysed by immunoblot. **g** EMSA using whole-cell extracts from TNFα- (10 ng/ml) stimulated AR42J WT and DKO cells. **h** Dexamethasone/EGF-treated AR42J WT and DKO cells were analysed for Sox9 and Hnf1b expression by RT-qPCR. *n* = 4. *p = 0.01. **i** Sox9 expression was analysed in AR42J DKO cells re-expressing κB-Ras2 by RT-qPCR. *n* = 4. *p = 0.036 (WT-DKO); *p = 0.025 (DKO + ctrl-DKO + κB-Ras2). **j** AR42J WT cells were stimulated with dexamethasone/EGF with or without BQU57 treatment and analysed for Sox9 and Hnf1b expression by RT-qPCR. Sox9: *n* = 5. *p = 0.0188; Hnf1b: *n* = 3. **k** DKO AR42J cells were treated with BQU57 and Sox9 expression was analysed by RT-qPCR. *n* = 3. *p = 0.04. **l** AR42J WT cells stably expressing RalA/B shRNAs were treated with TGFβ (10 ng/ml) and analysed for Sox9 expression by RT-qPCR. *n* = 3. **p = 0.0056; *p = 0.0356. **m** AR42J DKO cells stably expressing RalA/B shRNAs were analysed for Sox9 mRNA levels by RT-qPCR. *n* = 3. **p = 0.0023 (Sox9); **p = 0.0043 (Hnf1b). Source data are provided as a Source data file.

increased Ral-GTP levels, but did not alter NF-κB activity in resting or stimulated conditions (Fig. 8f, g, Supplementary Fig. 8e, f). Also, EGF-induced phosphorylation of Akt and Erk kinases was not affected by loss of κB-Ras (Supplementary Fig. 8g). AR42J cells can be induced to transdifferentiate into ductal cells in vitro by dexamethasone/EGF or TGFβ treatment[48,49]. Both AR42J κB-Ras DKO lines showed high expression of Sox9 and its target gene, Hnf1b (hepatocyte nuclear factor 1-beta)[50], already under resting conditions, which in the wild-type cells was only achieved after dexamethasone/EGF or TGFβ treatment, suggesting that κB-Ras knockout induces transdifferentiation in this system (Fig. 8h, l). Re-expression of κB-Ras2 in the DKO line rescued the effect, demonstrating that it is indeed due to κB-Ras deficiency (Fig. 8i). Importantly, upregulation of Sox9 and Hnf1b expression by AR42J cells upon stimulation of transdifferentiation, as well as the enhanced Sox9 expression in DKO AR42J cells, could be dampened by BQU57 treatment and Ral GTPase knockdown, demonstrating that Ral activation contributes to Sox9 induction in response to EGFR signalling (Fig. 8j–m, Supplementary Fig. 8h, i).

In summary, these results demonstrate that the absence of κB-Ras proteins promotes ADM through aberrant Ral signalling in acinar cells constituting a favourable environment for KRas$^{G12D}$-dependent carcinogenesis and preventing acinar regeneration after acute pancreatitis.

## Discussion

Here, we demonstrate a tumour-suppressive role for κB-Ras proteins in KRas$^{G12D}$-driven PDAC, and identify a relevant function of these proteins in regulating ADM during tumorigenesis and pancreatitis. We show that κB-Ras proteins control both tumour initiation through their role in acinar plasticity and progression through direct effects on proliferation of tumour cells. We furthermore reveal that deregulation of Ral activity through loss of Ral-GAP function underlies both of these phenotypes. With these findings, we identify the Ral GTPases as drivers of ADM and advance our understanding of processes driving early tumour development in the pancreas.

Heterozygous expression of KRas$^{G12D}$ in the murine pancreas is well characterised to initiate ADM and lock metaplastic cells in their duct-like state. However, this occurs with low penetrance and long latency. Secondary events are required for potent ADM induction and progression to carcinoma, which include additional genetic alterations or enhancement of the activities of wild-type and mutant KRas alleles through chronic inflammation or upregulation of growth factor signalling[51–55]. Chronic activation of EGFR also drives ADM in vitro and in vivo independently of KRas mutation[56,57]. To date, the PI3K/Akt and MAPK pathways have been suggested as the main drivers of ADM downstream of activated KRas[47]. Indeed, transgenic expression of constitutively active forms of PI3KCA/p110α and Akt1 drives ADM and cooperates with mutant KRas in PDAC development, and genetic deletion of PI3KCA/p110α prevents ADM in response to induction of acute pancreatitis[8,9,58–60]. Our results now reveal that Ral activity can also promote ADM during tumorigenesis and pancreatitis. Whether the different Ras effector pathways collaboratively drive ADM or represent redundant means to achieve metaplasia remains to be determined. Notably, while we did not observe alterations upon κB-Ras knockout in ADM cell-culture models, we detected continued low-level activation of Akt and MAPK signalling in persistent ADM lesions in pDKO mice in vivo. We thus cannot exclude the existence of regulatory crosstalk mechanisms between different Ras effector pathways in this scenario. Interestingly, in the case of pancreatitis, κB-Ras deficiency prevented acinar regeneration, suggesting that κB-Ras

deregulation and consequent enhancement of Ral activity not only lowers the hurdle for initial triggers to induce ADM, but also maintains ductal states by preventing redifferentiation to acinar phenotypes. This impact on regeneration has to our knowledge not been observed for chronically active PI3K/Akt and MAPK signalling. Ral inhibition should thus be considered for ongoing treatment efforts aiming at reprogramming PanIN or PDAC to normal pancreatic cell types such as acinar cells[61], and as a potential preventive option in cases of chronic pancreatitis. How downregulation of κB-Ras levels occurs in the pancreas remains a topic of ongoing research. In the patient samples analysed, KRas mutation status did not correlate with κB-Ras expression levels, suggesting that additional genetic alterations might be necessary.

Dedifferentiation of mature acinar cells generally requires activation of Sox9. We demonstrate that enhancing Ral activity through κB-Ras loss drives Sox9 expression in KRas$^{G12D}$-expressing acinar cells, while Ral inhibition prevents Sox9 induction in different systems assaying acinar transdifferentiation. Ral GTPases have previously been suggested to regulate Sox9 protein but not mRNA levels through unknown mechanisms during chondrogenic differentiation[62]. Our data rather suggest a transcriptional regulation of Sox9 by Ral activity; however, it is possible that stabilisation of Sox9 protein drives Sox9 mRNA expression through a feed-forward loop. Additional transcription factors that are involved in Sox9 regulation are NFATC1 and 4 (Nuclear factor of activated T cells, cytoplasmic 1 and 4), Pdx1 (Pancreatic and duodenal homeobox 1), Onecut1 (One cut homeobox 1) and the transcription activator BRG1 (Brahma-related gene-1), none of which has reported connections to Ral GTPase signalling[45,46,63–65]. Ongoing studies are thus addressing how Ral GTPases are involved in Sox9 control in the pancreas.

The role of Ral GTPases in PDAC has to date only been investigated in human cell lines. Here, knockdown of RalA reduces AIP in vitro and subcutaneous growth in vivo, while reduction of RalB impairs in vitro invasion and metastasis formation upon tail-vein injection[43,66,67]. Our results demonstrate that additional enhancement of active Ral levels is able to further boost AIP and invasion but not metastasis in the presence of constitutively active Ras. These findings are important for two reasons. First, relevance of loss of Ral-GAP activity has previously only been demonstrated for bladder cancer, where it promotes Ral activity and tumour invasion in cases without Ras mutations[68]. Our results now show a critical role of GAP control also in the presence of constitutive upstream signalling. As our data suggest a dosage-dependent role for κB-Ras, already a partial decrease of κB-Ras levels will likely impact cellular behaviour. Second, κB-Ras deficiency did not support the establishment of metastases in vivo although it enhanced invasion of both murine and human PDAC cells in in vitro assays. While it is possible that the necessary early euthanasia of pDKO/R mice prevented the detection of metastases that might occur at later stages, we consider it more likely that additional genetic triggers, besides KRas mutation, are required to initiate metastasis, which is then supported by Ral-dependent effects on cell migration and invasion.

Despite the described role of κB-Ras proteins in inflammatory signalling in macrophages[13], we did not detect the effects of κB-Ras deficiency on NF-κB activity in resting or stimulated pancreatic cells. From this, we conclude that κB-Ras-dependent regulatory processes are not relevant for the constitutive NF-κB activity observed in PDAC cells or involved in controlling stimulus-induced NF-κB activity in response to extrinsic inflammation. This suggests that the role of κB-Ras proteins in inflammatory signalling, which is mediated by interaction with the inhibitor protein IκBβ[13], is cell-type dependent. We hypothesise that this might in part depend on the presence or lack of

p65/c-Rel NF-κB heterodimers, as our observations hint that κB-Ras proteins might control the nuclear function of IκBβ[69]. Our findings therefore suggest that the increased pancreatic inflammation observed in pDKO/R mice is likely a secondary effect due to enhanced tumorigenesis and not intrinsically driven by κB-Ras-deficient acinar or ductal cells. Different extents of inflammation in pDKO/R resulting in variable degrees of cytokine levels that elicit NF-κB activation in surrounding cells thus likely underlie the observed variability of p65 phosphorylation in vivo. In summary, we conclude that the observed tumour-suppressive capabilities of κB-Ras proteins in the pancreas can largely be attributed to control of Ral GTPase activity. We expect that the here established mouse and cell line models, which allow us to specifically read out the effects of constitutively enhanced Ral activity while leaving other Ras-driven pathways intact, will be useful for the identification of targetable signalling events downstream of Ras/Ral activity in the future.

## Methods

**Cell lines and plasmids**. PDCs, Mia-Paca2 and Panc1 cells were cultured in RPMI medium containing 5% foetal bovine serum (FBS) and penicillin/streptomycin. AR42J and HEK293FT cells were cultured in DMEM medium containing 10% FBS, 1% L-glutamine and penicillin/streptomycin. Cells were treated with final concentrations of 1 μg/μl LPS O26-B6 (Sigma), 50 μM BQU57 (Sigma), 10 ng/ml TGFβ (Peprotech), 10 ng/ml TNFα (Peprotech), 10 ng/ml IL-1α (Peprotech) or IL-1β (Peprotech). Murine RalA S28N and RalA F39L mutants were cloned with an N-terminal 3× flag-tagged pcDNA3 vector via BamHI and NotI restriction sites. The Ral-binding domain of rat Sec5 (aa 1–99) and the Ras-binding domain of human Raf1 (aa 51–131) were cloned via BamHI and NotI restriction sites into pGex6p1 or pGex-4T1, respectively, for recombinant, IPTG-induced, expression of the protein in *E. coli* BL21DE3. Human κB-Ras2 was cloned into pBabe hygro vector via BamHI and SalI restriction sites. The sequence of murine κB-Ras2 was cloned into pCTAP vector via BamHI and XhoI restriction sites. The sequence of human or murine shRalA/B, as well as murine shSox9 or scramble shRNA oligonucleotides, was cloned into pLKO.1 puro vectors via AgeI and EcoRI restriction sites. shRalA/B sequences were designed to target both Ral GTPases simultaneously. All cloning procedures were performed using standard ligation protocols. Oligonucleotides are listed in Supplementary Table 2.

**Animal experiments**. Experimental animals were generated by crossing Pdx1-Cre[35] and LSL-KRas[G12D][5] with conventional κB-Ras1[23] and conditional κB-Ras2 mice. All mice were on a C57BL/6 background. Mice were housed in individually vented cages (IVC) containing nesting material. Constant ambient temperature (22 ± 2 °C), constant humidity (55% ± 10%) and a 12-h light/12-h dark cycle was provided. For glucose tolerance tests, 11-week-old mice were fasted for 6 h (water ad libitum) prior to intraperitoneal injection of 2 mg/g bodyweight D-glucose (in sterile water). Blood samples were taken from the tail vein just before and at several time points (15, 30, 60 and 120 min) after glucose injection. Blood glucose levels were determined with a standard glycosometer. Amylase levels were determined in serum from 11-week-old mice using a colorimetric Amylase activity Kit (Sigma) according to the manufacturer's protocol. Acute pancreatitis was induced by administration of 7 hourly intraperitoneal injections of cerulein (100 ng/g bodyweight in 0.9% saline) after a fasting period of 12 h (water ad libitum). Control animals received injections with 0.9% saline. Pancreata were analysed 9 h, 24 h, 48 h and 21 d after the first injection. All animal experiments were approved by the local animal use and care committee (LANUV) and the office of animal welfare of the University Clinic Münster.

**FACS analysis**. Pancreata were resected, minced into 2–4-mm small pieces in cold HBSS and washed two times with cold HBSS. Minced pancreata were digested in 30 mg/ml dispase I (Sigma) and 30 mg/ml collagenase IV (Worthington-Biochem) supplemented with soybean trypsin inhibitor (Gibco, 0.1 mg/ml) and DNaseI (20 μg/ml) at 37 °C for 60–80 min, carefully pipetting every 15 min. The single-cell suspension was washed two times with cold RPMI without phenol red, supplemented with 10% FCS. Cells were resuspended in RPMI containing 2% FCS and filtered through a sterile 40-μm cell strainer. Cells were surface-stained with MIC1-1C3 (Novus) and/or Mpx1 antibodies (gift from C. Dorrell, both 1:200, 15 min, on ice, dark). For intracellular stainings, cells were additionally fixed and permeabilized using the FoxP3/Transcription Factor staining Buffer or Intracellular Fix & Perm Sets (eBioscience). Antibodies used for intracellular staining were anti-Sox9 (D8G8H anti-rabbit Cell Signalling), anti-p-NF-kappaB p65 (S536 anti-rabbit, Cell Signalling, 3031), anti-p-TBK1/NAK (S172, D52C2, anti-rabbit, Cell Signalling), FITC goat anti-rat (BD Pharmingen) and Alexa Fluor 594 goat anti-rabbit (Invitrogen). For quantification of leucocytes, cells were blocked with anti-CD16/32 (93, eBioscience) (1:100, 10 min, ice, dark) and then stained with PE/Cy7 anti-mouse

CD45.2 (Clone 104, Biolegend) for 30 min. Flow cytometry analysis and sorts were performed on a FACSAria II using the FACSDiva version 6.1.3 and FACSuite version 1.0.6 software (BD Biosciences). Flow cytometry data were analysed using FlowJo version 10 (Treestar).

**Immunohistochemistry**. Pancreata were fixed in 4% paraformaldehyde and embedded in paraffin. In total, 5-μm sections were deparaffinised with xylene, rehydrated stepwise and stained with haematoxylin/eosin (H&E) or Alcian Blue/Nuclear Red (Biozol) according to standard procedures or the manufacturer's instructions. For immunostaining, antigen retrieval was performed using sodium citrate buffer (pH 6). Non-specific binding sites were blocked for 1 h in 2% donkey serum (Jackson Immuno Research) in PBS. Endogenous peroxidases were blocked with methanol:30% $H_2O_2$ (9:1) for 15 min. Slides were stained with primary antibodies overnight in a humidified chamber. Primary antibodies and dilutions used: anti-CK19 (1:100, Santa Cruz, sc-33111), anti-Amylase1 (1:100, Cell Signalling, D55H10), anti-Sox9 (1:100, Cell Signalling, D8G8H), anti-Cpa1 (1:100, Novus, AF2765), anti-phospho-p44/42 (1:200, Cell Signalling, Thr202/Tyr204), anti-Ki67 (1:100, Cell Signalling, 12202, clone D3B5) and anti-phospho-Akt substrate (1:100, Cell Signalling 9611). Slides were incubated with secondary antibodies in blocking buffer for 1 h. Secondary antibodies used: biotinylated anti-rabbit IgG (H + L) (1:250, BA-1100) and biotinylated anti-goat IgG (H + L) (1:250, BA-9500). For fluorescent immunostaining, the following antibodies were used: donkey-anti-rabbit-Alexa Fluor 488 (1:250, Jackson Immuno Research) and donkey-anti-goat-Cy3 (1:250, Jackson Immuno Research). Slides were mounted in ProLong Diamond Antifade Mountant with DAPI and analysed on a LSM 700 laser-scanning confocal microscope (ZEISS). Images were processed using Zeiss ZEN Black edition 2.3 SP1 software.

**In vitro ADM assay**. Pancreata were resected, minced in 5 ml of ice-cold HBSS and digested by adding 100 μl of 10 mg/ml collagenase P for 15 min at 37 °C. The digestion was stopped by adding cold HBSS containing 5% FBS and centrifugation at 17,800 × g for 2 min at 4 °C. After five washes in 5 ml of cold 5% FBS/HBSS, the cell solution was filtered through sterile 500-μm and 100-μm cell strainers, then layered on top of cold HBSS containing 30% FBS and centrifuged at 8900 × g for 2 min, 4 °C. Acini were resuspended in Waymouth's medium containing 20 μg/ml dexamethasone, 1 μg/ml soybean trypsin inhibitor, 25 ng/ml EGF and 100 ng/ml TGFα, with and without inhibitors, and mixed 1:1 (v:v) with neutralised rat tail collagen (2.5 mg/ml) and then plated on rat tail collagen- coated 24-well plates. After incubation for 1 h at 37 °C and 5% $CO_2$, warm Waymouth media containing 20 μg/ml dexamethasone, 1 μg/ml soybean trypsin inhibitor, EGF (25 ng/ml) and TGFα (50 ng/ml) with and without inhibitors was added on top of the gel. For lentiviral infection, isolated acini were resuspended in Waymouth's medium containing 20 μg/ml dexamethasone, 1 μg/ml soybean trypsin inhibitor, 25 ng/ml EGF and 100 ng/ml TGFα. Virus was harvested 48 h after transfection of HEK293FT using polyethylenimine (PEI) and mixed with the acini suspension in a ratio of 1:1 using polybrene (6 μg/ml f.c.). Cells were incubated for 1 h at 37 °C gently mixing the cells every 15 min followed by another 4 h of incubation at 37 °C. Acini were plated as described above. Fresh media was added on days 1 and 3. Samples were analysed on day 5. For viability analysis, collagen-embedded acini were incubated with 1 mg/ml Collagenase Ia (ThermoScientific) in HBSS for 30 min at 37 °C. The digestion was stopped by adding cold HBSS, and centrifugation was performed at 13,350 × g for 5 min. After two washes in 2 ml of cold PBS, cells were resuspended in 100 μl of PBS, stained with trypan blue and counted using a Neubauer chamber.

**Viral infection**. For retroviral and lentiviral transductions, HEK293FT cells were seeded at $4 × 10^6$ cells per 10-cm dish. The next day, cells were serum-starved for 1 h prior to transfection using PEI. For retroviral infections, 6 μg of pBabe and 6 μg of pCLEco vector, and for lentiviral infections, 6 μg of pLKO.1 puro vector, 5.4 μg of packing plasmid (psPAX2) and 0.6 μg of envelope plasmid (pCMV-VSV-G), were diluted in 360 μl of PBS, mixed with 48 μl of PEI (1 mg/ml stock) and incubated for 20 min at room temperature prior to addition to the cells. Virus was harvested 48 h after transfection to infect target cells using polybrene (4 μg/ml f.c.). Selection was started 36 h after infection.

**Cellular growth assays**. Cells were seeded in 24-well ultra-low-attachment plates at a density of 1000 cells per well in 500 μl of DMEM/F12 media supplemented with 1:50 B27, FGFβ (20 ng/ml), EGF (20 ng/ml), PenStrep and 1% FBS. In total, 100 μl of fresh media, including supplements, were added every 5 days. Colonies were counted on day 14. For adherent growth analysis, 500 cells per well were seeded in a white cell-culture-treated 96-well plate. Cell viability was measured daily using CellTiter-Glo (Promega) according to the manufacturer's instructions.

**Colony-formation assay**. For colony-formation assays, 1000 cells were resuspended in 500 μl of 1% SeaPrep™ Agarose (Lonza) in RPMI supplemented with 10% FBS and seeded on 12-well plates coated with 1% agarose/PBS. After solidification, the agarose layer was overlaid with 0.5 ml of RPMI containing 10% FBS. Every third day, 100 μl of fresh media were added. After 14 days, colonies larger than 20 cells were scored.

**Invasion assay.** A 8-µm pore transwell was overlaid with matrigel (BD Biosciences) diluted 1:10 with serum-free RPMI-1640 medium and dried overnight. Matrigel was rehydrated for 2–3 h in serum-free medium. In total, $2.5 \times 10^5$ cells in 100 µl of serum-free RPMI-1640 were seeded per transwell. RPMI-1640 containing 10% FBS was added to the deep wells. After 18 h, cells were fixed with 4% PFA, permeabilised with 100% methanol, stained with Giemsa solution and counted.

**RNA preparation and RT-qPCR.** RNA was prepared using TRIzol™ (Life Technologies) following the instructions provided by the manufacturer. cDNA was generated using the RevertAid RT Reverse Transcription Kit (Thermo Fisher Scientific) from 200 to 1000 ng of RNA per reaction. SYBR Green-based quantitative PCR was performed using Luna Universal qPCR Master Mix (NEB). Data collection and analysis were performed using 7300 System SDS Software version 2.3. RT-qPCR primers are listed in Supplementary Table 2.

**CRISPR-/Cas9-mediated knockout.** gRNA sequences were cloned in px458 (Addgene pSpCas9(BB)-2A-GFP (PX458) #48138) or px459 (Addgene pSpCas9 (BB)-2A-Puro (PX459) #48139) vectors as reported previously[70]. gRNAs were designed to delete whole exons by placing two double-strand breaks (see Supplementary Figs. 4d and 8c). Panc1, Mia-Paca2 and AR42J cells were transfected with two plasmids (one gRNA in px458 and the second in px459) simultaneously using JetPrime according to the manufacturer's instructions. Cells were sorted 24 h after transfection and subsequently treated with puromycin for 36 h. After recovery, single-cell clones were isolated via limited dilution. Successful deletion was identified via a screening end-point PCR as well as RT-qPCR. Oligonucleotides and primers are listed in Supplementary Table 2.

**GST-Sec5 and Raf1 PD.** A total of $1–3 \times 10^6$ cells were seeded per 10-cm dish. The next day, cells were serum-starved for 24 h. Cells were stimulated with EGF (100 ng/ ml f.c.) for the indicated time points, and extracts prepared using Ral lysis buffer (50 mM Tris, pH 7.5, 100 mM NaCl, 4 mM $MgCl_2$, 2 mM EGTA and 1% Triton X-100). GST-Sec5 (rat, aa 1–99) as well as GST-Raf1 (human, aa 51–131) were expressed recombinantly in *E. coli* BL21DE3 and purified via affinity (Glutathione Sepharose) and gel-filtration chromatography. GST-Sec5 and Raf1 were then rebound to Glutathione Sepharose. Sepharose beads containing 20 µg of recombinant effector protein were incubated with the cell lysates for 25 min rotating at 4 °C. Beads were washed three times with Ral lysis buffer and analysed for Ral-GTP or Ras-GTP levels by immunoblot using anti-RalA, anti-RalB and anti-Ras antibodies.

**Immunoblotting.** Proteins separated by sodium dodecyl sulfate polyacrylamide gel electrophoresis were blotted on polyvinylidene fluoride membrane using a semidry system. Membranes were blocked using Licor Blocking buffer (TBS). The following antibodies were used: κB-Ras (antibody raised against recombinant human κB-Ras1), NKIRAS2 (Proteintech), Ras (Cell Signalling, 3965), RalA (BD Biosciences, clone 8/ralA), RalB (Origene, OTI2C4), Dab2 (BD Biosciences, 52/p96), α-tubulin (Santa Cruz, sc-23948), GAPDH (Proteintech, 60004-I-Ig), p53 (Santa Cruz, sc-1312), Smad4 (Santa Cruz, B8, sc-7966), p44/42 MAPK (Cell Signalling, 137F5), p-p42/p44 MAPK (Cell Signalling, Thr202/Tyr204), p-Akt (Santa Cruz, T308, sc-16646), p-Akt (Cell Signalling, Ser473, 4060, D9E), p-Akt (Cell Signalling, Thr308, 4056, 244F9), phospho-Smad2 (Cell Signalling, 3108T), Akt (Cell Signalling, 9272) AND Smad2/3 (Cell Signalling, D7G7). All primary antibodies were diluted 1:1000. Secondary antibodies were purchased from LI-COR: IRDye®800CW (α-mouse), IRDye®680RD (α-rabbit) and diluted 1:10,000. Bands were visualised on a Licor Odyssey CLx system (Image Studio Software version 5.2 LI-COR).

**EMSA.** Double-stranded oligonucleotides were labelled using $P^{32}$-γ-ATP and T4 polynucleotide kinase. Whole-cell extracts (Bäuerle buffer: 20 mM TRIS, pH 8, 350 mM $NaCl_2$, 20% glycerine, 1 mM $MgCl_2$, 0.5 mM EDTA, 0.1 mM EGTA and 1% NP-40) were incubated with labelled oligonucleotides in 20 mM HEPES, pH 7.9, 60 mM KCl, 4% Ficoll, 2 mM DTT,100 ng/ml bovine serum albumin and 100 ng/ ml Poly dI-dC in 20-µl reactions for 30 min at room temperature. Samples were run on 5% acrylamide/TBE gels at 200 V/26 mA. Sequences for oligonucleotides are listed in Supplementary Table 2.

**Patients and clinicopathologic data.** In total, 81 formalin-fixed, paraffin-embedded PDAC tissue samples from 61 patients were included in the study. Detailed clinicopathological data were retrieved from the respective pathology reports/clinical records, and are summarised in Supplementary Table 1. The use of human tissue samples was approved by the ethics committee of the University of Münster (Approval Number 2015-102-f-S).

**Immunohistochemistry of PDAC tissue microarray (TMA).** The construction of PDAC TMAs has been described previously[71]. All tissue samples were collected for the purpose of histopathologic diagnosis and anonymised for the use in this study. Informed consent was therefore not needed to be obtained. This was approved by the ethics committee of the University of Münster (Approval Number 2015-102-f-S). About 4 µM thick slides were cut and stained using anti-κB-Ras antibody (1:50 dilution) as previously described[71]. In brief, sections were deparaffinised in

xylene and rehydrated through graded ethanol at room temperature. Incubation with the primary antibodies was performed for 30 min at room temperature. After washing, the sections were incubated with biotinylated secondary antibodies. Immunoreactions were visualised using a 3-amino-9-ethylcarbazole as a substrate (Ventana Optiview DAB IHC detection Kit, Ref: 760-700, Germany). κB-Ras expression was evaluated by a pathologist blinded with respect to clinical data.

**Statistics and reproducibility.** Statistical analysis was conducted using Prism Software (GraphPad Prism version 8). Statistical tests used are mentioned in the respective figure legends. If not stated otherwise, all *n* numbers represent independently performed experiments, and statistical tests are two-sided. Where representative experiments are shown, experiments have been performed at least two times independently. The results demonstrated in Fig. 4e and Supplementary Figs. 4b, 5d and 5e have been obtained twice independently. Experiments shown in Fig. 4f–l have been performed three times independently. Data presented in Fig. 6a, b, d and Supplementary Figs. 2c and 7a, b have been obtained from three biologically independent animals in three independently performed experiments.

**Reporting summary.** Further information on research design is available in the Nature Research Reporting Summary linked to this article.

## Data availability
The data that support the findings of this study are available from the corresponding author upon reasonable request. Source data are provided with this paper.

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

## Acknowledgements

We thank Claudia Schlosser, Tanja Grote-Westrick and Thorsten König for excellent technical assistance. Our thanks also go to Dr. Craig Dorrell for providing anti-Mxi1 antibodies and Dr. Thomas Wirth for Pdx1-Cre mice. pSpCas9(BB)-2A-GFP (PX458) was a gift from Feng Zhang (Addgene plasmid # 48138; http://n2t.net/addgene:48138; RRID:Addgene_48138). pSpCas9(BB)-2A-Puro (PX459) V2.0 was a gift from Feng Zhang (Addgene plasmid # 62988; http://n2t.net/addgene:62988; RRID:Addgene_62988). Our thanks go to Dr. Haoqiang Ying and Dr. Ronald A. DePinho for technical advice. This work was supported by grants of the German Research Foundation (DFG; OE531/2-1 and OE531/2-2) and the Innovative Medical Research (IMF) Initiative of the University of Münster (OE121701) to AO.

## Author contributions

S.B., L.K., L.H.A, L.J., A.B., N.S. and A.O. performed the experiments and analysed the data. K.S. conducted all histological analyses and grading. E.W., S.G. and M.M. gave technical support and conceptual advice. A.O. designed the study, and A.O. and S.B. wrote the paper.

## Competing interests

The authors declare no competing interests.
